# Central Nervous System and Ocular Manifestations in Puumala Hantavirus Infection

**DOI:** 10.3390/v13061040

**Published:** 2021-05-31

**Authors:** Nina Hautala, Terhi Partanen, Anna-Maria Kubin, Heikki Kauma, Timo Hautala

**Affiliations:** 1Medical Research Center, PEDEGO Research Unit, Department of Ophthalmology, Oulu University Hospital and University of Oulu, 90029 Oulu, Finland; nina.hautala@oulu.fi (N.H.); anna-maria.kubin@oulu.fi (A.-M.K.); 2Research Unit of Internal Medicine, Department of Internal Medicine, Division of Infectious Diseases, Oulu University and Oulu University Hospital, 90014 Oulu, Finland; terhi.partanen@ppshp.fi (T.P.); heikki.kauma@ppshp.fi (H.K.); 3Research Unit of Biomedicine, University of Oulu, 90029 Oulu, Finland

**Keywords:** hantavirus, central nervous system, encephalitis, visual disorders, intraocular pressure, myopic shift

## Abstract

Puumala hantavirus (PUUV), carried and spread by the bank vole (*Myodes glareolus*), causes a mild form of hemorrhagic fever with renal syndrome (HFRS) called nephropathia epidemica (NE). Acute high fever, acute kidney injury (AKI), thrombocytopenia, and hematuria are typical features of this syndrome. In addition, headache, blurred vision, insomnia, vertigo, and nausea are commonly associated with the disease. This review explores the mechanisms and presentations of ocular and central nervous system involvement in acute NE.

## 1. Introduction

Hemorrhagic fever with renal syndrome (HFRS) is a zoonosis caused by Puumala hantavirus (PUUV). The condition is common in Northern Europe, and a number of cases have been found in Central and Western Europe [1,2,3]. This acute febrile condition, nephropathia epidemica (NE), can be endemic in rural environments; individuals exposed to rodents present the highest risk of infection. Bank vole (Myodes glareolus) is the reservoir host of PUUV, which infects humans via inhalation of urine, feces, or saliva aerosols.

NE patients may suffer from an acute onset of high fever, headache, oliguria, and blurred vision up to two weeks after exposure to infectious bank vole aerosols. Patients commonly complain of central nervous system (CNS) or ocular-related symptoms during their acute infection [1,4]. These associated symptoms, together with acute kidney injury (AKI), oliguria, hematuria, proteinuria, and thrombocytopenia, often lead to a precise preliminary diagnosis even before the serological test results are available. While most ocular and CNS-related symptoms are reversible, serious complications such as pituitary hemorrhage and encephalitis have been reported [5,6,7,8]. Progress in understanding the clinical and biological features of blurred vision and hantavirus CNS involvement has been made recently.

## 2. CNS and Acute NE

### 2.1. CNS Symptoms

Ahlm et al. reported a high frequency of CNS symptoms in acute NE caused by PUUV [9]. Headache (96%), insomnia (83%), vertigo (79%), nausea (79%), vomiting (71%), and anxiety (38%) were common complaints. In another recent study by Hautala et al., 51 of the 58 (87%) patients described CNS symptoms such as headache, nausea/vomiting, dizziness, or light sensitivity [6]. Alexeyev et al. investigated 811 NE patients’ records, paying particular attention to data regarding neurological manifestations [10]. They found that the most common symptoms were headache (97%), blurred vision (40%), and vomiting (31%). These case series demonstrate that the CNS is commonly affected in NE.

Rare and serious CNS symptoms have been described. For instance, in the study conducted by Hautala et al., two patients experienced a complete and sudden loss of vision among the 58 cases [6]. The vision loss resolved spontaneously some minutes later. Both patients had a pituitary hemorrhage in their acute brain magnetic resonance imaging (MRI), whilst the other also developed long-term hormonal deficiency. In addition, cases of spontaneous pituitary bleeding have been reported in association with PUUV or Dobrava hantaviruses [5,7,11,12]. Alexeyev et al. reported cases of meningism, epileptic seizures, urinary bladder paralysis, and cerebral hemorrhage in their NE patient cohort [10]. Seizures, confusion, somnolence, and vision loss, mainly among young male patients, have also been reported [8,13,14]. These reports highlight the potential of the acute NE to cause serious and possibly irreversible CNS complications.

### 2.2. PUUV Encephalitis

Acute NE may rarely present symptoms and findings consistent with viral encephalitis criteria such as altered mental status or generalized seizures [15]. Moreover, abnormalities of brain parenchyma on neuroimaging and on electroencephalography have been reported [8,10,13,14]. Based on personal unpublished clinical experience, most NE encephalitis patients recover well without long-term sequelae. Cases of permanent disability related to CNS symptoms, however, have been described [8].

Genetic susceptibility to PUUV encephalitis has been suggested in a recent study by Partanen et al. They found a variant in the Toll-like receptor 3 (TLR3) gene leading to compromised receptor activity and an abnormal interferon response [8]. The described TLR3 p.L742F variant appeared to be enriched in the Finnish population (www.sisuproject.fi, accessed on 20 May 2021). It seems possible that PUUV encephalitis can be more common in populations with genetic susceptibility, especially in conjugation with endemically high NE frequencies. Similar genetic mechanisms have been described in encephalitis caused by herpes simplex virus 1 and 2 (HSV1 and HSV2), for example [16,17]. 

TLR3 receptor recognizes double-stranded RNA (dsRNA), which is a typical intermediate product of viral replication. TLR3 activation triggers antiviral responses such as interferon production, NK cell activation, and inflammation [18,19,20]. The heterozygous novel TLR3 p.L742F variant was observed in two of the seven unrelated NE encephalitis patients (29%, *p* = 0.0195) [8]. The authors generated TLR3-deficient P2.1 fibrosarcoma cell line and SV40-immortalized fibroblasts from patient skin expressing the p.L742F variant and tested them functionally. A low poly(I:C)-stimulated cytokine induction in the TLR3 p.L742F P2.1 cell line was recognized. In addition, the cytokine production was impaired in TLR3 L742F/WT patient SV40-fibroblasts. They concluded that a possibility of TLR3 deficiency should be considered in patients with PUUV hantavirus encephalitis. Furthermore, the possibility of a similar mechanism should be considered in rare cases of encephalitis as described in Dobrava hantavirus HFRS encephalitis in addition to Sin Nombre, Andes, and New York hantavirus HCPS encephalitis [21,22,23,24].

### 2.3. Blood-Brain Barrier Permeability Disorder

Acute NE is characterized by severely disturbed tissue permeability [1,25]. It has been suggested that the CNS symptoms in acute NE may arise from permeability disorders of the Blood-Brain barrier. This view is supported by the obvious fluid retention and tissue edema seen in the oliguric phase of the disease. The elevated CSF protein concentration is most obviously caused by leakage from the circulation [6]. The increased permeability may at least partly explain CNS symptoms in NE. Mechanisms leading to this barrier integrity disorder, however, are not well understood. These complex biological events causing endothelial cell dysfunction have been reviewed by Hepojoki et al. [26]. On the other hand, the possibility of direct viral invasion and PUUV infection of the CNS cells has been considered. It seems possible that the integrity of the Blood-Brain barrier or blood contamination of the CSF during lumbar puncture may also affect the results of CNS virological evaluation. We assume, however, that the PUUV may indeed invade the CNS. We also believe that the events associated with permeability disorder are at least partly responsible. 

### 2.4. Analysis of Cerebrospinal Fluid

Cerebrospinal fluid (CSF) samples were collected from 41 consecutive acute NE patients [6]. The CSF white cell count was elevated in 12 patients (mean 8.1 × 10^6^/L, range 4–35 × 10^6^/L) compared to normal values (0–5 × 10^6^/L). The CSF protein concentration was elevated in nearly half of the patients (mean 987 mg/L, range 519–3265 mg/L, normal <450 mg/L). Several patients had an elevation in both their white cell count and protein concentration. The CSF neopterin concentration was high, especially in those who suffered from low thrombocyte count, high plasma creatinine concentration, and in those who required prolonged hospitalization [27]. This finding supports the possibility that a monocyte/macrophage lineage can be active in the CNS, particularly in patients with a severe form of NE [28]. Ahlm et al. measured interleukin-1, interleukin-6, and TNF-a CSF concentrations [9]. Although these markers were not elevated, the presence of PUUV in the CNS can be assumed to stimulate acute inflammation, which in turn can participate in the development of the symptoms.

Intrathecal antibody production against PUUV can activate during acute NE. Half of the patients were positive for PUUV IgM antibodies in the CSF in a series of acute NE patients [6]. The patients with a positive CSF PUUV IgM finding were more often males, and they had a high plasma creatinine value (*p* < 0.001) compared to those with a negative CSF PUUV IgM [13]. In addition, high PUUV IgG levels were found in these acute NE CSF samples [6]. High PUUV IgG titers were also measured in serum samples. In 10 patients in whom both serum and CSF PUUV IgM end-point titers were obtained, the serum/CSF ratio was lower compared to that typically observed in CSF samples. The CSF PUUV IgM was positive in patients with elevated CSF inflammatory cells and an elevated CSF protein concentration. The CSF pleocytosis, increased protein concentration, or presence of PUUV IgM can be associated with a high plasma creatinine concentration. These results suggest that patients with a severe NE and generalized tissue permeability disorder may develop significant CNS involvement. The PUUV IgM finding, in turn, may indicate the presence of the virus in the CNS [29]. It is thought that the size of the IgM may not facilitate the crossing of the Blood-Brain barrier even in the presence of permeability disorder triggered by PUUV.

### 2.5. PUUV in the CNS

PUUV has been demonstrated in the CNS in a limited number of patients. First, post-mortem samples of a male patient, who died during acute NE, was shown to have PUUV in his CNS [5,6]. The post-mortem examination showed that the patient’s pituitary gland was enlarged and hemorrhagic. Histological tissue sections confirmed that the patient’s pituitary gland was necrotic, hemorrhagic, and infiltrated with polymorphonuclear cells. In addition, his dural and subdural cavity was also infiltrated with inflammatory cells. Furthermore, the endothelial cell layer in the pituitary gland was positive for the Puumala virus N–antigen. Further analysis of the serial tissue samples for cell type-specific markers (chromogranin for neuroendocrine cells, CD31 for endothelial cells, and S-100 for glial cells) and the Puumala virus N protein showed that either the chromogranin-positive endocrine cells or the CD31-positive endothelial cells were infected by PUUV. The presence of PUUV RNA was also found using reverse-transcriptase PCR (RT-PCR) in the patient’s CNS samples.

The presence of PUUV RNA has also been demonstrated in an acute CSF sample [30]. A 58-year-old male had suffered from fever and oliguria for three days whilst developing slight edema on his lower extremities. He had also experienced photophobia, impaired vision, and irritation of his eyes. The patient’s acute phase CSF sample was negative for erythrocytes, and his white blood cell count was 1 × 10 × 10^6^/L. The patient’s CSF protein (418 mg/L) and glucose (4.3 mmol/L) concentrations were within a normal range. The PUUV RNA was positive both from the CSF and blood samples. The PUUV sequences found from the blood and CSF (Genbank accession number EF625237) had a high identity compared to those obtained from the Finnish PUUV strains. This case was the only patient with a positive CSF PUUV RNA among the study participants [5]. It is possible to witness PUUV viremia only during the very early days of NE. Moreover, severe thrombocytopenia may not allow the collection of the CSF sample at the viremic phase of the illness. 

### 2.6. Electroencephalography

Ahlm et al. performed electroencephalography (EEG) on acute PUUV NE patients [9]. The EEGs were abnormal in three out of twenty patients. One patient had a slight episodic slowing over the right temporal region, and another patient had a mild, generalized slowing in the EEG finding. These changes had normalized in the follow-up. One patient suffered an episode of generalized tonic-clonic seizure; severe EEG changes were recorded 9 days after the onset of symptoms. The EEG showed a mild generalized slowing and frequent episodes of symmetric irregular delta activity with a precentral maximum. Seven days later, some frontal intermittent rhythmic delta activity remained, but five weeks later, the EEG had normalized. Hautala et al. reported that the EEG was normal in all 33 hospitalized NE patients [6,9]. However, patients with major CNS complications such as seizures, confusion, or vision loss had abnormalities in their EEGs [13]. 

### 2.7. Pituitary Gland Involvement in Acute NE

Pituitary injury associated with NE has been described [5,6]. Some of the patients with pituitary bleeding have experienced an immediate and complete loss of vision in acute NE. This loss of vision, which is thought to be caused by optic chiasm compression, may last for minutes and, in most cases, it resolves spontaneously. In a recent case series, young male patients were at an elevated risk of developing pituitary injury or other forms of serious CNS complication during acute NE [13]. Some, but not all, patients with pituitary bleeding develop a lasting hormonal deficiency [31]. Partanen et al. analyzed the NE patients 4 to 8 years after the acute infection. In their study, none of the patients developed late-onset hypopituitarism even though the pituitary gland can be affected during acute NE. The pituitary size had diminished to normal levels, and hormonal levels were mainly within normal limits. In another study, 30 of the 54 NE patients exhibited abnormalities in their thyroid and/or gonadal axis in acute NE [32]. 9 of the 54 patients continued to have findings of hormonal deficit after a median follow-up of 5 years. Furthermore, evidence suggesting a possibility of autoimmune polyendocrinopathy and hypophysitis associated with NE has been presented [33]. Features of hormonal deficiencies associated with hantaviruses have been recently reviewed by Bhoelan et al. [34].

### 2.8. Brain Imaging

Radiological findings associated with NE have been reviewed recently [35]. Hautala et al. performed brain MRI on 45 acute NE patients [6]. The imaging was normal in 24 of the 45 patients. The imaging results were judged to be unspecific without an obvious connection to the acute NE in 15 patients. No intracranial bleeding outside of the pituitary was noted. In six patients, abnormality of the pituitary gland was found. Two patients had a pituitary hemorrhage. Uneven distribution of the contrast media in two patients, suspicion of pituitary edema in one patient, and a pituitary cyst in one patient were found. None of the patients had imaging evidence of encephalitis. Each patient was evaluated for his or her pituitary gland 4 to 8 years after the acute NE episode [31]. The size of pituitaries had significantly diminished in all patients. 

## 3. Ocular Manifestations and NE

### 3.1. PUUV and Diversity of Ocular Features

The ocular symptoms are often the hallmark feature in NE [4,9]. These symptoms, in association with signs of acute infection, should lead to a suspicion of NE, at least in endemic regions. Reduced vision is the most common ocular feature in NE. Transient myopia or myopic shift, changes in ocular dimensions, acute glaucoma attacks, decrease in intraocular pressure (IOP), anterior uveitis, lid edema, conjunctival chemosis, conjunctival hemorrhages, retinal edema and hemorrhages, diplopia, pupillary defect, and anisocoria have also been detected previously in patients with NE [4,6,9,36,37,38,39,40,41,42,43,44]. These findings, however, are often based on case reports or studies involving a limited number of patients at different stages of the disease with a variation in severity. These previous studies have also reported some controversial results of the changes in intraocular pressure or the presence of anterior uveitis [36,38,39,44,45], for example. 

Reduced visual acuity is a highly characteristic finding in the acute NE. A majority (87%) of the patients suffered from reduced visual acuity in a study of 46 NE patients [4]. In agreement with these results, over half of the patients, 54%, complained of blurred vision in a case series of 37 NE patients [44]. Altogether, 70% of patients had experienced at least one ocular symptom [4]. Light sensitivity, foreign-body sensation and discomfort, and orbital pain were common ophthalmic complaints. CNS symptoms such as headache, nausea/vomiting, or dizziness were also common (87%) among these patients [6]. Frontal headache or periocular pain have been reported in 76% of NE patients in a study by Kontkanen et al. [44]. Photophobia has been described in three (20%) of the fifteen patients with acute NE, which was the only ocular manifestation of NE in two patients [37,38]. In another study, 11% of the NE patients complained of photophobia [44]. The occurrence of photophobia, as well as other ocular symptoms, may vary in different stages of NE. The profile of symptoms may also depend on the severity of the infection as well as the extent of documentation of the symptoms [6].

### 3.2. Changes in Ocular Dimensions and Refractive Power of the Lens

Transient myopia in NE has been first described by Saari et al. [37,39]. Saari and Luoto reported a myopic shift in 8 out of 15 patients (53%) [38]. They suggested that relaxation of the lens zonules and a forward-change in the lens position would have caused a change in refraction. This myopic shift was also confirmed in 41% of the patients with NE in a study by Kontkanen et al. [44]. They found a narrowing of the anterior chamber in 93% and a thickening of the lens in 87% in the eyes of the NE patients. In agreement with these results, transient myopia, forward movement of the anterior diaphragm, and a thickening of the crystalline lens were detected in a case report of a 35-year-old woman [46]. This phenomenon has been addressed even more clearly in a study demonstrating that 78% of the patients had a myopic shift and 88% a thickening of the lens [4]. The mean myopic refraction change was −0.7 D (range −0.25 to −4.5 D, *p* < 0.001) in a total of 68 eyes of 34 patients (74%). Refraction remained unchanged in 17 eyes (18%), and in seven eyes (8%), refraction changed into the hyperopic direction (mean +0.4 D, range +0.25 to +0.75 D). The changes in refraction power were analogous between the right and left eye in an individual patient, suggesting the systemic nature of the ophthalmic changes.

Shallowing of the anterior chamber was noted in 93% of the patients suffering from myopic shift [42]. According to another Finnish study, shallowing of the anterior chamber in 64% of NE patients explained the myopic change only partly [4]. Moreover, deepening of the anterior chamber with simultaneous myopic change in the acute phase of NE was noted in 27% of the patients. These findings do not support the hypothesis that the change in the anterior chamber depth is a major mechanism of myopic shift [42]. Thickening of the lens is another potential explanation for myopic shift [4,42,45]. The thickening of the lens was reported in a majority, 82% and 87% of all NE patients, reported with a simultaneous myopic refraction change in the two largest studies, which supports this hypothesis [4,44]. The increased thickness of the lens evokes a myopic change by steepening the anterior and posterior curvatures of the lens. It is speculated that the refractive index of the lens may be affected due to changes in the osmolarity of both the lens and the aqueous humor. Simultaneously, with the thickening and the possible forward movement of the lens, the length of the vitreous cavity tended to be shallower, and no significant change in the axial length was noted during acute NE [4]. The thickening of the lens has been thus considered the key mechanism in contributing to the myopic change in acute NE in contrast to the previous study [42]. 

The symptoms and ocular findings of NE are transient; it seems possible that examination of refraction at earlier or later time points could lead to a different conclusion. Furthermore, the change of refraction towards a myopic direction does not necessarily lead to decreased visual acuity in patients with an acute phase of NE. In contrast, the myopic shift at the acute phase of NE may even momentarily improve the visual acuity of hyperopic patients when not wearing eyeglasses.

### 3.3. Intraocular Pressure (IOP) in NE

Contradictory results of IOP level in patients with NE have been published previously [4,37,38,39,43,44,45,47,48]. There are some case reports of acute glaucoma attacks associated with NE [37,38,39,48,49,50] and several reports of decreased IOP during the acute phase of NE [4,43,45]. Cho et al. have reported a single case of bilateral acute angle-closure glaucoma in Korean hemorrhagic fever, a very severe form of HFRS caused by the Hantaan virus [48]. PUUV may multiply in capillary endothelial cells, which may, in turn, increase capillary permeability and lead to leakage of osmotically active material and erythrocytes. This can be detected as interstitial hemorrhages in the kidneys, heart, lungs, and pituitary glands [51,52]. It is thus possible that increased capillary permeability causes edema and hemorrhages in the ciliary body during acute NE.

The short-duration increase in IOP, described as acute angle-closure glaucoma, is possibly caused by edema and hemorrhage in the ciliary body, its anterolateral rotation, relaxation of the zonules, and the anterior movement of the lens [37]. The angle-closure in their study was caused by posterior synechiae due to anterior uveitis in one patient and shallowing of the anterior chamber due to relaxation of the lens zonules and transient forward movement of the lens in another two patients. In a study by Saari and Luoto, only one out of fifteen NE patients developed acute angle-closure glaucoma [38]. Recently, a case report of a fleeting increase in IOP up to 30 mmHg in the right eye and 24 mmHg in the left eye of a 32-year-old man associated with NE was published [49]. Optical coherence tomography and ultrasound biomicroscopy were used to detect ciliochoroidal effusion and iridocorneal angle-closure as the mechanisms of the bilateral angle-closure glaucoma. The rarity of the angle-closure glaucoma cases among the number of patients evaluated during acute NE could be explained by the temporary nature of the blockage and spontaneous healing in most cases.

In contrast to the reports above, several studies have demonstrated a decrease in IOP during acute NE. In a previous study by Kontkanen et al., an average of 1.9 mmHg decrease in IOP was noted in 66% of the cases [44]. A decrease of the IOP has also been reported by Kontkanen et al. in a single patient with NE [46] and in a study of five patients with an acute hantavirus infection by Mehta et al. [47]. In agreement with these results, 88% of the total of 46 NE patients had an average of 4.5 mmHg decline in IOP during the acute phase, compared to the follow-up evaluation after the recovery [4]. No acute angle-closure attacks were detected among NE patients in this study. According to these results, decreased IOP appears to be a general characteristic of NE. Transient angle-closure attacks, however, may infrequently occur in selected NE patients. It is known that aqueous flow from the ciliary processes into the posterior chamber is principally the result of metabolic pump activity in the ciliary epithelia. Diminished aqueous formation and filtration in the ciliary body due to damage of capillary endothelial cells could be a logical explanation for the reduced lOP. The analogy between decreased filtration of the ciliary body and the oliguric phase of NE is evident. 

### 3.4. Eyelid and Conjunctival Changes

Bilateral eyelid edema, conjunctival chemosis, injection, and hemorrhages have been described in three of the fifteen (20%) NE patients [37,38]. These changes were suggested to be caused by the leakage of the conjunctival capillaries reported in the fluorescein angiography. Another study revealed almost 40% presentation of lid edema, conjunctival hyperemia, chemosis, or subconjunctival bleeding, which were diagnosed in 27%, 11%, and 4% of the study patients, respectively [44]. Kontkanen and Puustjärvi have also reported conjunctival hemorrhages in the case of one female patient with NE [46]. Conjunctival chemosis was noted to some extent in 87% of the eyes in a previous prospective study, in which only one patient suffered bilateral subconjunctival bleeding [4,49]. The acute phase of NE causes an increase in tissue permeability and interstitial hemorrhages. These may result from the leakage of erythrocytes, which has been described in many organs [53]. This could also explain the frequency of conjunctival chemosis and subconjunctival bleeding in NE patients.

### 3.5. Uveitis in NE 

Uveitis has been diagnosed in a few NE patients only. Signs of acute uveitis have been documented in 7% and 14% of the eyes during the acute NE by Saari and Kontkanen et al., respectively [37,44]. These cases of uveitis, however, have resolved without treatment. In a prospective study of 46 NE patients, no cases of anterior or posterior uveitis were detected [4]. The increase in tissue permeability in the capillary endothelial cells of the ciliary body may have triggered a condition resembling uveitis, rather than actual inflammation of the uvea in these cases, because of the typical spontaneous recovery of uveitis during acute NE.

### 3.6. Retinal Changes

Unilateral retinal edema and hemorrhages were detected in one of the thirty-seven patients (3%) [44]. Accordingly, Saari and Luoto have described retinal hemorrhages in one patient with subsequent acute glaucoma and anterior uveitis [38]. Mehta et al. have reported dot and blot hemorrhages in the macula and streak hemorrhages of the optic disc in two patients in a study including five patients with hantavirus infection [47]. No cases of retinal hemorrhages were recorded in a study involving 46 patients with NE by Hautala et al. [4].

Retinal pigment epithelial swelling at the posterior fundus concomitant with the mild anterior uveitis and decrease in vision in connection with acute nephritis has also been described previously in a Finnish case report [40,46]. A recent case report described, for the first time, self-limiting posterior necrotizing retinitis and vasculitis possibly caused by the Hantaan River species of hantavirus supported by intraocular production of anti-hantavirus antibodies [4,54].

## 4. Conclusions

CNS and ocular symptoms appear to be very common in acute NE [6]. Most patients suffer from symptoms consistent with eye and CNS involvement. In addition, recent data provide evidence that the PUUV is capable of invading and infecting the CNS [27]. The inflammation caused by the virus and the permeability disorder leading to alteration in the Blood-Brain barrier integrity are also significant. 

Nearly all patients recover from NE without permanent consequences on any organ. Only very rare cases of encephalitis, pituitary gland injury, or other serious CNS complications have been described [5,13,31]. The first evidence of genetic predisposition to these serious consequences has been presented. We suggest that infection caused by PUUV or other hantaviruses should be considered whenever febrile meningitis or encephalitis of an unknown infectious origin is suspected, depending on geographical regions endemic for the PUUV or other hantaviruses [55,56]. It is also evident that serious CNS complications, such as pituitary hemorrhage, should be considered, at least in those patients who have suffered from vision loss and/or long-term disability associated with NE [5,6].

Ophthalmic findings, such as reduced vision, myopic shift, shallowing of the anterior chamber, and changes in IOP are common in NE. The timing of the ophthalmic evaluation during the acute febrile and oliguric phase may affect the ocular findings and thus explain at least some variation of the results. The presentation of most ophthalmic findings is bilateral, suggesting the reflection of the systemic nature of the underlying infection. Not all ocular manifestations, however, can be explained by ocular mechanisms only, and the impact of overall pathophysiological alterations of acute NE on ophthalmic changes is understandable. 

## Data Availability

All data is presented in references.

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
