# Peer review of "Central Nervous System and Ocular Manifestations in Puumala Hantavirus Infection"

_viruses, 2021, doi:10.3390/v13061040_

Round 1

Reviewer 1 Report

“Central Nervous System and Ocular Manifestations in Puumala Hantavirus infection” by Hautala and co-writers brings together the present knowledge of central nervous system (CNS) involvement and ocular manifestations during acute Puumala hantavirus (PUUV) infection. The article is a comprehensive review of mechanisms and presentations of these common manifestations.

Acute PUUV infection is characterized by abrupt start of high fever, vomiting, nausea, headache, back and abdominal pains and reduction of urinary output. Transient phase of blurred vision is a typical complain as well. Typical symptoms and findings are well described in the current literature. By now, there is, however, only one review article about CNS manifestations in hantaviral diseases (Bhoelan et al. Hypopituitarism after Orthohantavirus Infection: What is Currently Known? Viruses. 2019;11:340), and no review articles that focus on ocular manifestations.

The present article is well written and especially chapter 3 that discuss about ocular manifestations is an awaited addition to the current literature. Ocular findings are not familiar to most clinicians that treat patients with acute PUUV infections. The review is rather long, and some shortening is recommended.

I have only minor comments:

Row 29. In medical literature, the term “renal failure” is not recommendable any more. Use instead: “acute kidney injury”. Add also “proteinuria” since proteinuria is almost as common finding as hematuria in acute PUUV infection.

Row 110. The elevation of neopterin in cerebrospinal fluid and its association to more severe PUUV infection is an interesting finding. Could you give a short explanation why neopterin is increased during acute PUUV infection. Is it increased nonspecifically in viral CNS infections?

Author Response

Reviewer 1.

  1. We thank the reviewer for the positive comments
  2. Some shortening is recommended.

Response: We have condensed selected parts and sentences. Revised text appears in red font.

  1. Term “renal failure” is not recommendable anymore.

Response: Term “renal failure” is replaced with “acute kidney injury (AKI)”. “Proteinuria” is also added to the list of common findings in NE.

  1. Could you give a short explanation why neopterin is increased during acute PUUV infection?

Response: We have added a new reference (Eisenhut et al) explaining the role of neopterin.

Reviewer 2 Report

General. The article provides information on “Central Nervous System and Ocular Manifestations in Puumala Hantavirus Infection”, which is common in Norhtern Europe. The authors base their research on data from occurrence of the disease in Finland, but not provide any summarized data about annual incidence of the disease. We suggest not to economize in description of the disease and their symptoms. The main aim of the every review article to unite and systematize all data , which was not done properly. The article is lack information about neural signs and ocular manifestations related to other viral diseases and clinical signs, caused by infection.

It is difficult to understand what is significance in this study: an unique symptoms affecting eye, irreversible changes in eye cause by viral infection, helpful information for making diagnosis, information which is usable in treatment of such patients etc. Many sentences make me feel incompleteness. The reader doesn’t need to be puzzled what author mean.   Whether these symptoms are specific for hantaviral infections or common for other hemorrhagic and viral diseases? How often these changes neurologic, ocular , without known etiology) are seen in patients without acute PUUV infection in common population?

In this form the article will not provide clear information to the reader, which would be not only ophthalmologists, but also related to other professions.

In many sections no control groups, no comparison with patients of the same age/cohort were not shown. These doesn’t allow to understand and evaluate whether these changes are linked to the HUUV viral infection or not. No additional information illustration the same changed didn’t shown. In case when this information is not available, it also should be added to the text

Line 11-12. “ Puumala hantavirus (PUUV) carried and spread by bank vole causes a mild form of hemorrhagic fever with renal syndrome (HFRS) called nephropathia epidemica (NE).” We suggest to add the world zoonosis, or human. The sentence should be corrected

Introduction.

The order of sentences in the introduction should be better organized. The full nomenclature of the virus should be added. Generally, the introduction should provide more details about nephropathia epidemica and complication of the disease. The aim/purpose of the article should be shown. Acute kidney injury (AKI) and thrombocytopenia Acute kidney injury (AKI) and thrombocytopenia are the main clinical sings of nephropathia epidemic, isn't it? If so the author should pay attention to the main clinical sings and after that to rare ones. We presume (as we seen in literature) Hanta viral infection usually asymptomatic, or only febrile. This information should be added to the section.

.

Line 13. “ ja” Is it correct?

Line 14. “ In addition, headache, blurred vision, insomnia, vertigo, and nausea are common complaints.” I suggest to add “ of the disease” to the end of the sentence.

Line-15-16. The sentence should be re-written. The virus damage several organ systems. Please, provide information on affected systems of organs and frequency of other symptoms in HUUV affected patients.

Line 22. "nephropathia epidemica"- definition and short description of syndrome (?) has to be added. Proportion of patients with neural symptoms among total number of NE patients should be shown also. Whether the number of NE patients with neurological symptoms is small or not?

Line 22-23. The sentence should be rewritten. The authors speak about two things: exposure to the rodents and about the virus, and syndromes/symptoms of the disease. It should be written clearer. Whether the symptoms appeared simultaneously, or begin from one type of clinical signs and continued with the second type? It is not understood from the sentence. This sentence should be written after the following sentence.

Line 27. The patients commonly compline. We suggest to add "with acute NE" to the sentence before commonly.

Line 28” their acute infection”. What authors mean under the “ their acute infection”. How it is proven? It should be described in more details.

Line28-29.” These associated with disturbed vision, renal failure, oliguria, hematuria and thrombocytopenia often leads to correct diagnosis even before the serological test results are available.” The phrase "preliminary diagnosis" should be added. Whether it is spoken on differential diagnosis of this disease (NE)?

Lines 30-31. First and the second parts of the sentence have no connection. Did the authors write about reversible vs non-reversible symptoms or serious vs light symptoms? Not clear.

Lines 32-33. “ Progress in understanding the clinical and biological features of blurred vision and hantavirus CNS involvement has been made recently”. What do you mean in this sentence and what progress had been made? No citation.

The purpose/aim of this review article should be illustrated here.

Line 37-39. Should be mentioned that reference 13 describe non-PUUV NE related symptoms.

Lines 39. “ were common complaints” of what? What did the author want to illustrate in this sentence? Please, make changes.

Line 39: "in another…" – is this article supportive or not?

Lines 37-44. I suggest to begin the paragraph from the last sentence.

Lines 41: "eight hundred eleven" should be eight hundred and eleven.

Line 46. The word "have" should be changed to "had"

2.2. Section. The name and content of the section are poorly correspond one to another.

Line 59: CSF – the full name of abbreviation should be mentioned here.

Line 61: reference 14 presented twice instead of one.

Line 61. " These symptoms and findings may fulfill the encephalitis criteria [20]". Did you mean that these symptoms were caused by viral infection? If so, the sentence should be completed.

Line 68. " to be enriched" the word is not suitable for this meaning. (frequency?)

Line 71." HSV1 and HSV2"- full name of viruses have to be added.

Line 73." TLR3 receptor activation" order of words should be changed

Line 74. with particular importance in the CNS [23,24]". It should be better explained

Line 75. " The TLR3 engagement may also stimulate inflammation and NK cell activation"- how it happen? The citation is absent.

Line 90. “Acute NE is characterized by severely disturbed tissue permeability”. Usually, when hemorrhagic viral disease takes place, it is spoken about endotheliosis, and as consequence, increased permeability of blood vessels, which lead to edema and hemorrhages. The correction should be done in this part of the text.

Lines 90- 91. “It has been suggested that the CNS symptoms in acute NE may arise from permeability disorder of the blood-brain barrier.” Citation is lacking.

Line 94. “increased permeability” of what? It should be completed.

Line 101: Citation is absent

Line 101. “We assume, however, that the PUUV may indeed invade the CNS”. This sentence should be placed into conclusions.

Lines 102-103. “We also believe that the events associated with permeability (of blood vessels?) disorder are at least partly responsible.” The thought is unclear. Why did the author believe that the events associated?

Section 2.4. The title not illustrate the context and should be changed

Line 109: "several" – how many patients?

Line 115: sentence is poorly phrased

Line 118. “Intrathecal antibody production against PUUV can activate during acute NE.” Did the authors mean “can be activated”?

Lines 118-133. Citation is lacking or authors shown write that it’s a new data. Is the two articles referred to the same cohort?

Line 131. Please, check whether the word ”size” is really suitable here. (level, amount?)

Line 133: citation is absent.

Section 2.5. whole section is poorly written.

Line 135: "limited number of patents" – how many out of total?

Line 139. Presence of “polymorphonuclear cells” should mentioned that it is usually illustrate bacterial but not a viral infection.

Line 140. “infiltrated with inflammatory cells.” Which specific cell were found? It should be signed.

Line 140-141. Which method (I suppose immunochemical of FISH) was used? Citation is absent.

Lines 135-146 and 147-158. The authors write about two were different methods of antigen identification. It should be separated into two sections and described better.

Lines 152-153. “The PUUV RNA was positive for PUUV from the CSF and blood.” The sentence Method is not described. Citation is absent.

Line 154. “cerebrospinal fluid lysate” was signed as “isolation_source“ for EF625237 in GenBank. Please, make changes in the text accordingly.

Lines 157-158. Here authors describe pathogenesis. It should be written in another part of the article.

Line 160. “Ahlm et al. completed electroencephalography (EEG) to acute NE patients.” whether the authors mean “ performed” or not?

Section 2.7. this section should be more general to PUUV and non-PUUV NE. should compare to non-PUUV NEs.

Lines 172-174. “Some of the patients with pituitary bleeding have experienced an immediate and complete loss of vision in acute NE.” It wasn’t shown how pituitary bleeding was detected.

Lines 173-174. “immediate and complete loss of vision in acute NE.’ the section has name” Pituitary gland involvement in acute NE”. Authors should change or title, or put this data in another place of the article.

Line 188. Hautala et al. completed brain MRI to 45 acute NE patients [6]. The word “completed” is not suitable in this context.

Lines 189-190. “The imaging results were judged to be unspecific without an obvious connection to the acute NE in 15 patients.” Which group 15 patients belong: to the total(45) or unspecific(24)?What did the authors want to show/prove?

Lines 194-195. “These patients were followed for their pituitary gland 4 to 8 years later”. Function? Hormone production? Size?

Section 3.

Line 199. Lack of citation.

Line 208: please elaborate what are the main controversy between studies.

Lines 214-217. These clinical signs are not belonged to “Ocular symptoms and reduced visual acuity” as signed in the title.

Section 3.3. The section is poorly written

Lines 269-273. “PUUV may multiply in capillary endothelial cells, which may in turn increase capillary permeability and lead to leakage of osmolar active material and erythrocytes. This can be detected as interstitial hemorrhages in kidneys, heart, lungs, and pituitary gland [50,51]. It is thus possible that increased capillary permeability causes edema and hemorrhages in the ciliary body during acute NE.” it is not belonged to this section. It is pathogenesis and should be described or in separate section, or in the introduction section.

Line 295. “According to these results, decreased IOP appears to be 295 a general characteristic of NE

Line 297." It is know" – citation is lacking

Line 302. Lack of citation.

Line 304. “injection”? What did the authors mean?

Line 317. “Uveitis has been diagnosed in a few NE patients only.”

Line 336. “caused by hantavirus” it can be proven post mortem only. According to the citation [3]- this case was not fatal, wasn’t it? It should be proven.

Author Response

  1. We suggest not to economize in description of the disease and their symptoms.

Response: we have revised the introduction and added new references.

  1. How often these changes neurologic, ocular, without known etiology) are seen in patients without acute PUUV infection in common population?

Response: Hantaviruses most often cause a recognizable condition. Discussion of differential diagnostic measures may not be possible due to limited focus of this manuscript.   

  1. In many sections no control groups, no comparison with patients of the same age/cohort were not shown.

Response: we have added references to each individual study. Most studies do not have control groups. Instead, they describe findings of NE patients only.

  1. The order of sentences in the introduction should be better organized.

Response: we have partly reorganized the introduction

  1. Line 13. “ ja” Is it correct?

Response: revised

  1. Whether the number of NE patients with neurological symptoms is small or not?

Response: we explain the numbers in the revised text

  1. "eight hundred eleven" should be eight hundred and eleven.

Response: revised

  1. CSF – the full name of abbreviation should be mentioned here

Response: revised

  1. reference 14 presented twice instead of one.

Response: revised

  1. " These symptoms and findings may fulfill the encephalitis criteria [20]"

Response: an explanation has been added

  1. " to be enriched" the word is not suitable for this meaning.

 Response: an explanation for genetic properties of Finnish population has been added

  1. " HSV1 and HSV2"- full name of viruses have to be added.

Response: full names appear in revised text

  1. " The TLR3 engagement may also stimulate inflammation and NK cell activation"- how it happen?

Response: a reference has been added

  1. Please, check whether the word ”size” is really suitable here.

Response: we believe that “size” is correct term

  1. Sections are poorly written.

Response: some rephrasing has been done

  1. “It has been suggested that the CNS symptoms in acute NE may arise from permeability disorder of the blood-brain barrier.” Citation is lacking.Line 94. “increased permeability” of what? It should be completed.

Response: references has been included

  1. Section 2.4. The title not illustrate the context and should be changed

Response: selected titles are rephrased

  1. Line 139. Presence of “polymorphonuclear cells” should mentioned that it is usually illustrate bacterial but not a viral infection. Line 140. “infiltrated with inflammatory cells.” Which specific cell were found? It should be signed.

Response: all essential information that is available has been provided

  1. Line 140-141. Which method (I suppose immunochemical of FISH) was used? Citation is absent. Lines 135-146 and 147-158. The authors write about two were different methods of antigen identification. It should be separated into two sections and described better. Lines 152-153. “The PUUV RNA was positive for PUUV from the CSF and blood.” The sentence Method is not described. Citation is absent. Line 154. “cerebrospinal fluid lysate” was signed as “isolation_source“ for EF625237 in GenBank. Please, make changes in the text accordingly.

Response: methods of the original work are not provided in order to limit the length of the chapter

  1. Line 160. “Ahlm et al. completed electroencephalography (EEG) to acute NE patients.” whether the authors mean “ performed” or not?

Response: “completed” has been replaced with “performed”

  1. Section 2.7. this section should be more general to PUUV and non-PUUV NE. should compare to non-PUUV NEs.

Response: the text mainly describes PUUV; non-PUUV is only briefly described

  1. Lines 172-174. “Some of the patients with pituitary bleeding have experienced an immediate and complete loss of vision in acute NE.” It wasn’t shown how pituitary bleeding was detected.

Lines 173-174. “immediate and complete loss of vision in acute NE.’ the section has name” Pituitary gland involvement in acute NE”. Authors should change or title, or put this data in another place of the article.

Response: mechanism of vision loss is briefly explained in the revised manuscript

  1. Lines 189-190. “The imaging results were judged to be unspecific without an obvious connection to the acute NE in 15 patients.” Which group 15 patients belong: to the total(45) or unspecific(24)?What did the authors want to show/prove?Lines 194-195. “These patients were followed for their pituitary gland 4 to 8 years later”. Function? Hormone production? Size?

Response: the manuscript is rather long - details of the studies can be found in the references

  1. Lines 214-217. These clinical signs are not belonged to “Ocular symptoms and reduced visual acuity” as signed in the title.Line 297." It is know" – citation is lacking. Line 302. Lack of citation. Line 304. “injection”? What did the authors mean? Line 317. “Uveitis has been diagnosed in a few NE patients only.”

Response: literature on ocular findings in NE is limited and all essential sources of information are cited in the manuscript. Term “conjunctival injection” is widely used in ophthalmology and does not require an explanation.
